# Marine-Based Candidates as Potential RSK1 Inhibitors: A Computational Study

**DOI:** 10.3390/molecules28010202

**Published:** 2022-12-26

**Authors:** Mousa AlTarabeen, Qosay Al-Balas, Amgad Albohy, Werner Ernst Georg Müller, Peter Proksch

**Affiliations:** 1Department of Basic Medical Sciences, Faculty of Medicine, Aqaba Medical Sciences University, Aqaba 11191, Jordan; 2Institute of Pharmaceutical Biology and Biotechnology, Heinrich-Heine University, Universitätsstrasse 1, 40225 Düsseldorf, Germany; 3Department of Medicinal Chemistry and Pharmacognosy, Faculty of Pharmacy, Jordan University of Science & Technology, Irbid 22110, Jordan; 4Department of Pharmaceutical Chemistry, Faculty of Pharmacy, The British University in Egypt, El-Sherouk City, Cairo 11837, Egypt; 5The Center for Drug Research and Development (CDRD), Faculty of Pharmacy, The British University in Egypt, El-Sherouk City, Cairo 11837, Egypt; 6Institute of Physiological Chemistry, University Medical Center of the Johannes Gutenberg University Mainz, 55128 Mainz, Germany

**Keywords:** lymphoma L5178Y cell line, RSK1 N-terminal kinase, molecular docking, manzamine/es

## Abstract

Manzamines are chemically related compounds extracted from the methanolic extract of *Acanthostrongylophora ingens* species. Seven compounds were identified by our research group and are being characterized. As their biological target is unknown, this work is based on previous screening work performed by Mayer et al., who revealed that manzamine A could be an inhibitor of RSK1 kinase. Within this work, the RSK1 *N*-terminal kinase domain is exploited as a target for our work and the seven compounds are docked using Autodock Vina software. The results show that one of the most active compounds, Manzamine A *N*-oxide (**5**), with an IC_50_ = 3.1 μM, displayed the highest docking score. In addition, the compounds with docking scores lower than the co-crystalized ligand AMP-PCP (−7.5 and −8.0 kcal/mol) for ircinial E (**1**) and nakadomarin A (**7**) were found to be inferior in activity in the biological assay. The docking results successfully managed to predict the activities of four compounds, and their in silico results were in concordance with their biological data. The β-carboline ring showed noticeable receptor binding, which could explain its reported biological activities, while the lipophilic side of the compound was found to fit well inside the hydrophobic active site.

## 1. Introduction

More than 8000 new marine natural products (MNP) were recorded between 2001 and 2010 [1]. This number has grown significantly since then. Interestingly, numerous marine leads which are now under clinical trial are promising, and several of these agents are likely to reach the market in the coming years [2,3,4]. Six of ten marine metabolites approved by the FDA are anticancer agents, while three are marine-sponge-derived. Manzamines are a unique class of alkaloids present in marine sponges and possess a fused tetra- or pentacyclic ring system which is linked to a β-carboline moiety. Manzamines were initially reported from the Indo-Pacific sponge *Acanthostrongylophora ingens* [5]. Since then, more than 80 manzamine congeners have been isolated, including ircinal A [6], ircinal E [7],^,^ nakadomarin A [8], manzamine A [9], 12,28-oxamanzamine E [10], zamamiphidin A [11], zamamidine C [12], manzamine J *N*-oxide [13], and acantholactone [14]. Manzamine derivatives have attracted scientific interest due to their bioactivity potential, including antimalarial [9], anti-inflammatory, antiviral [15], anti-atherosclerotic [16], antimicrobial [10], proteasome-inhibitory [17,18], and cytotoxic [19] effects. Moreover, various molecular targets have been elucidated as valid targets for manzamines, namely, vacuolar ATPases [20], glycogen synthase kinase-3 (GSK-3), and cyclin-dependent kinase 5 (CDK5) [21]. Furthermore, Mayer A.M.S. et al. [22] outlined for the first time that manzamine A inhibited a 90 kDa ribosomal protein kinase S6 (RSK1) as they tested a group of 30 protein kinases. Moreover, it was found in this study that RSK kinase assays demonstrated a 10-fold selectivity in the potency of the same compound in vitro against RSK1 versus RSK2. In addition to that, the divergent binding and selectivity of manzamine A toward the two isoforms was supported by their computational docking experiments. According to these experiments, it has been found that the RSK1-manzamine A (*N*- and *C* termini) complexes appear to have stronger interactions and preferable energetics contrary to the RSK2-MZA ones. Additionally, they proposed that manzamine A binds to the *N*-terminal kinase domain of RSK1 rather than the *C*-terminal. RSK kinases consist of two functional kinase catalytic domains: the *C*-terminal kinase domain (CTKD) classified as the calcium and calmodulin-regulated kinases (CamK) family and phosphorylated by ERK1/2, and the *N*-terminal kinase domain (NTKD), which is classified as a protein kinase A, G, and C (AGC) family member [23]. RSK1 is a 90 kDa ribosomal S6 kinase which belong to vertebrate family of cytosolic serine–threonine kinases that contain four homologous isoforms, namely, RSK1-4. They act downstream of the ras-ERK1/2 (extracellular-signal-regulated kinase 1/2) pathway [23]. RSK1 is found in the lungs, kidneys, pancreas and brain (cerebellum and microglia) [23,24,25]. During our ongoing search targeting bioactive secondary metabolites from marine sources [7,26] guided by the study that has been published by Mayer A.M.S. et al. [22], we examined the Indonesian marine sponge of *Acanthostrongylophora ingens* collected at Ambon (Indonesia) in October 1996 [7].

The methanol extract of *Acanthostrongylophora ingens* has afforded seven manzamines, namely, ircinal E, manzamine A, 8-hydroxymanzamine, manzamine F, manzamine A *N*-oxide, 3,4-dihydromanzamineA *N*-oxide, and nakadomarin A (Figure 1) [7]. Within this work, these manzamine derivatives, previously isolated and characterized [7], were docked within the active site of the RSK1 *N*-terminal kinase domain of murine lymphoma L5178Y cell line.

## 2. Materials and Methods

The crystal structure of the RSK1 *N*-terminal kinase domain (NTKD) was downloaded from Protein Data Bank (PDB) under the PDB code 2Z7Q with a resolution of 2.00 Å. Autodock Vina [27] was used to perform molecular modeling in a cubic grid box with 25 Å sides centered on the co-crystalized ligand phosphomethylphosphonic acid adenylate ester (AMP-PCP) with exhaustiveness of 16. The protein and ligands were prepared as reported earlier [28,29]. In short, the protein was prepared using PyMOL software by stripping water molecules, adding hydrogens, and maintaining one side chain in case of duplicate sidechain amino acids. Ligands (Figure 1) were obtained from PubChem when possible as 3D structures, and were minimized and converted to standard pdbqt format using PyRx software. Docking was performed using Autodock Vina and the obtained nine poses were analyzed and arranged according to their docking score.

## 3. Results

Seven marine-derived compounds, as well as the co-crystalized AMP-PCP ligand, were docked in the active site of the RSK1 *N*-terminal kinase domain (NTKD). The AMP-PCP ligand was docked to the active site as a validation step. The aim of this was to confirm that the software was able to predict the correct pose of the AMP-PCP ligand. In addition, the docking score of AMP-PCP ligand was used as a guide to evaluate the docking scores of the tested ligands. Docking software protocol was accepted when an RMSD less than 2 Å was obtained between the docked pose and the crystal pose. The RMSD was found to be 1.472 Å using the DockRMSD server [30]. Docking results of the seven test compounds as well as their biological activity are shown in Table 1.

## 4. Discussion

Validation of the docking procedure was performed by redocking the AMP-PCP co-crystalized ligand and comparing the predicted pose to the crystal structure pose (Figure 2). In general, if the RMSD between both structures was less than 2 Å, the docking validation was accepted. The docking software was able to predict the correct pose with an RMSD of 1.472 Å and a docking score of −8.1 kcal/mol.

The docking results of the rest of the compounds are shown in Table 1. Based on these results, there is a good correlation between the reported IC_50_ and the docking scores for most of the compounds. Docking scores were able to segregate the compounds into active and inactive compounds. Two compounds showed docking scores weaker than the AMP-PCP ligand (−8.1 kcal/mol): ircinial E (1) and nakadomarin A (7). Docking scores for these two compounds were −7.5 and −8.1 kcal/mol, respectively. These two compounds showed weak or no activity in the biological assays. It worth mentioning here that these two compounds were missing the β–carboline ring compared with the other five compounds 2–6. This shows the importance of this part of the molecule for both the biological activity and docking score.

The remaining five compounds (2–6) showed a relatively small activity range in terms of IC_50_ (2.8–4.1 μM) and docking score (−9.8 to −10.6 kcal/mol). For example, Manzamine A (2) showed an IC_50_ of 3.3 μM and a docking score of −9.9 kcal/mol. Adding a hydroxyl group at position 8 forms 8-Hydroxymanzamine A (3), which has an improved activity (IC_50_ of 3.0 μM) as well as improved docking score (−10.2 kcal/mol). The binding modes of both compounds were very similar, except for the extra hydroxyl group, which formed a hydrogen bond with D205 and a weaker hydrogen bond with K94. In addition to hydrogen bonds, both compounds formed several hydrophobic interactions with hydrophobic residues in the active site, including L68, L194, V76, F150 and L144. These data suggest that the presence of the 8-hydroxyl group helps increase the cytotoxicity of the compound, as well as its docking score (Figure 3).

Revisiting the results portrayed in the Table 1, one can see that docking was not able to completely differentiate between compounds 2 and 6 in terms of activity. It is well-known that molecular modeling programs are rough indicators used to explain real results. However, these results greatly support the already published article by Mayer et al., which suggests that these compounds, which are derived from manzamine, are good candidates for RSK1. In addition, this study suggests a potential binding mode between this class of compound and RSK1, which will be useful for the design of further inhibitors.

In conclusion, manzamine derivatives could be good lead compounds for potent inhibitors of RSK1. Molecular modeling techniques have helped in predicting a possible target when docking studies have been conducted on RSK1. The results produced from this work have shown a good correlation between the real IC50 and the docking scores against RSK1. The β–carboline ring has been proven to be an important moiety, and its existence has contributed to manzamine’s activity. These results will guide any future efforts in designing manzamines and their semi-synthesized derivatives as potential RSK-1 inhibitors.

## Figures and Tables

**Figure 1 molecules-28-00202-f001:**
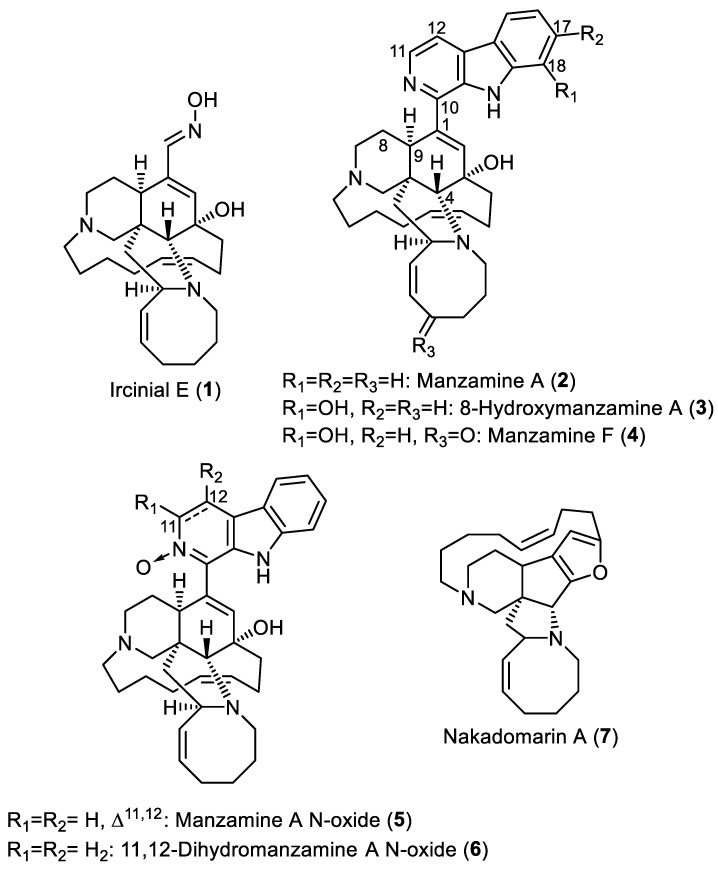
The chemical structures of the methanolic extract of *Acanthostrongylophora ingens* used in the in silico docking study.

**Figure 2 molecules-28-00202-f002:**
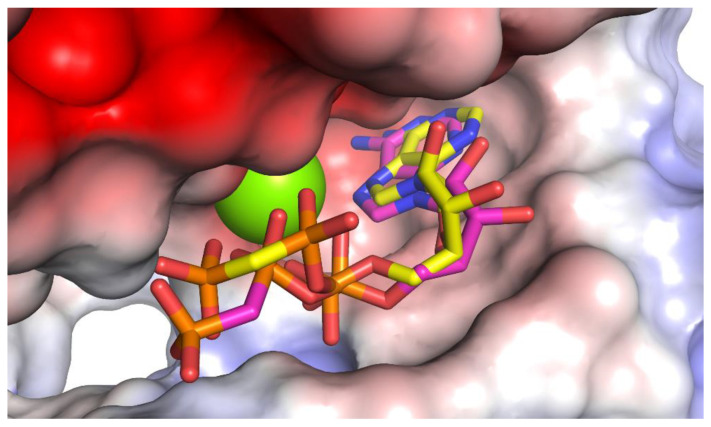
Validation of docking procedure by redocking of AMP-PCP ligand (yellow). The docking pose predicted (pink) is relatively similar to the crystal structure with an RMSD of 1.472 Å. Magnesium ion in the active site is shown as the green sphere.

**Figure 3 molecules-28-00202-f003:**
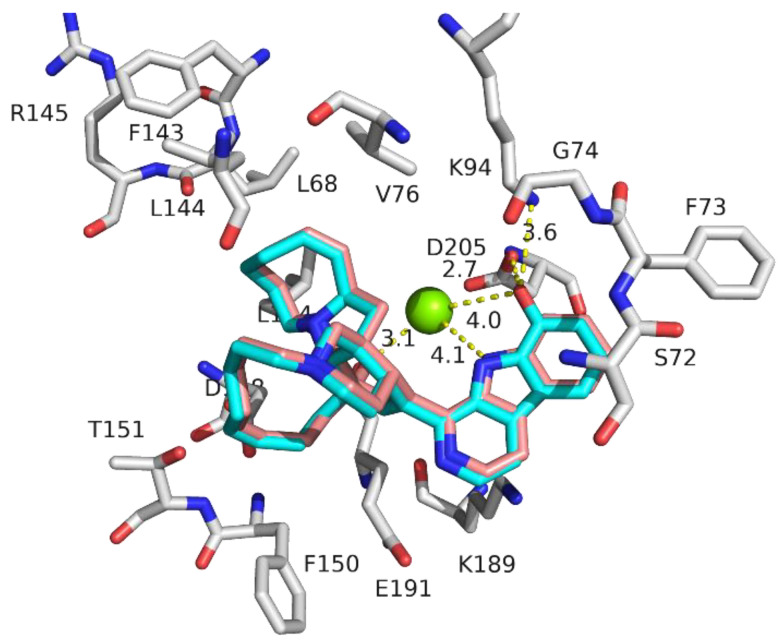
Three-dimensional representation of the docking pose of manzamine A (2, salmon) and 8-Hydroxymanzamine A (3, blue) within the active site of RSK1 N-terminal kinase domain (NTKD). Hydrogen bonds and interactions with Mg^+2^ for the β–carboline part of the molecule are shown as yellow–dotted lines. Hydrophobic interacting residues are shown and form a hydrophobic pocket in the active site.

**Table 1 molecules-28-00202-t001:** The docking scores and biological activity of a series of manzamine alkaloids with natural kinase substrate 2Z7Q.

Name	IC_50_ (µM)	Docking Score (kcal/mol)
Ircinial E (1)	21.7	−7.5
Manzamine A (2)	3.3	−9.9
8-Hydroxymanzamine A (3)	3	−10.2
Manzamine F (4)	4.1	−10.3
Manzamine A *N*-oxide (5)	3.1	−10.6
3,4-Dihydromanzamine A *N*-oxide (6)	2.8	−9.8
Nakadomarin A (7)	-	−8.0
AMP-PCP Ligand		−8.1

## Data Availability

Not applicable.

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
