# Peer review of "Marine-Based Candidates as Potential RSK1 Inhibitors: A Computational Study"

_molecules, 2022, doi:10.3390/molecules28010202_

Round 1
Reviewer 1 Report
The manuscript of the article entitled "Marine based candidates as potential RSK1 Inhibitors, a Computational Study" by Mousa AlTarabeen, Qosay Al-Balas, Amgad Albony, Werner Ernst Georg Müller and Peter Proksch should be interesting for readers of Molecules and as it extends the current level of knowledge. I have following comments and requirements: For all the compounds discussed, it is necessary to present their structural formulas. The form of the results presented in the text are not providing an easy survey. The calculated values for each compound including previously published results must be presented in the form of synoptic tables. It is necessary to present the results of the molecular modelling in the form of figures showing the interactions of the active site enzymes with the molecules of inhibitors. The present manuscript should be acceptable for publication in the Molecules after major revision.
Author Response
Dear Reviewer,
Thank you for your valuable comments. We addressed your comments and suggestions thoroughly. The major revision has been done. Please refer to the original manuscript as it was done by "Track changes" option in word. Also, All changes are highlighted in yellow in the manuscript. our detailed response to your comments can found as attached files.
Your Sincerely,
Mousa AlTarabeen

Reviewer 2 Report
This is an interesting and potentially useful work on the simulated docking capabilities of manzamine derivatives to the Nt-domain of RSK1 protein kinase, of biomedical importance (i.e. in strategies for cancer control). The work, although clear-cut, shows a significant number of points in the text that would require either at the level of typing or grammar/ linguistic construction mistakes/ or inappropriate particles, or either to clarify the meaning of some terms of the text. The following are different points or lines of the text as examples for such requirements, that should be revised.
i.e. : -At the Abstract, lines 24+25 (have showed/has showed: better substitute the second by “display”?) -lines 25-27 (excessive use of the terms “least” and “less”; merge them?) -At the Introduction, as in lines 36-37 (“… is very strong …”, better use “very promising” or equivalent) -line 48 (“… have been revealing …” ?) -lines 52 and 56 (on the use of “… it found”; it has been found?) -lines 61-62 (“… it is phosphorylated by ERK1/K2 …” Nothing else in the sentence?) -line 66 (“RSK1 found in …” , better use “RSK1 has been found …?). -line 75 (better remove “which”?) … The whole text should be checked for such kinconsistences/mistakes. The text also suffers the use of distinct inappropriate terms, as in: -line 20, “full” : probably better remove it because is doubtful that could be said that such compounds have actually been “fully characterized” (probably only “partially”). -line 32: why not include “manzamine/es” within the keywords? -lines 84-86 and 90-91, why citing a whole reference there and not simply its reference number, and place the whole one at the references list, at the back? -lines 26, 96 and others: about the use if the term “co-crystallized ligand”, why not use there its specific name, like for the other ligands in Table 1? Anyway, the meaning of this term and origin should be clearly explained. -Also, clarify whether the term EC50 used in Table 1 is such, or whether is IC50, frequently mentioned along the text.
Regarding the selected references to illustrate the large number of marine natural products of biomedical interest and that have been or still are under clinical trials, it looks a bit strange to select the ones restricted until 2010 (12 years until today), and not add o substitute it by more updated one, including ref. 2, which leads to a link to conference in which the list was updated until 2019. Probably, this issue could be better explained easily.
Author Response
Dear Reviewer,
Thank you for your valuable comments. We addressed your comments and suggestions thoroughly. The major revision has been done. Please refer to the original manuscript as it was done by "Track changes" option in word. Also, All changes are highlighted in yellow in the manuscript. Please could you find our detailed response to your comments as attached file.
Your Sincerely,
Mousa AlTarabeen

Round 2
Reviewer 1 Report
The revised manuscript can be published in Molecules in the present form.
Author Response
Thanks for your review.